# The Preparation of N-Doped Titanium Dioxide Films and Their Degradation of Organic Pollutants

**DOI:** 10.3390/ijerph192315721

**Published:** 2022-11-25

**Authors:** Yanyan Dou, Yixuan Chang, Xuejun Duan, Leilei Fan, Bo Yang, Jingjing Lv

**Affiliations:** 1School of Energy and Environment, Zhongyuan University of Technology, Zhengzhou 450007, China; 2Department of Resources and Environment, Zunyi Normal College, Zunyi 563006, China; 3People’s Government of Donganggezhuang Town, Luanzhou City, Tangshan 063000, China

**Keywords:** photocatalysis, N-doped films, visible light, titanium dioxide

## Abstract

N-doped TiO_2_ films supported by glass slides showed superior photocatalytic efficiency compared with naked TiO_2_ powder due to them being easier to separate and especially being responsive to visible light. The films in this study were prepared via the sol–gel method using TBOT hydrolyzed in an ethanol solution and the nitrogen was provided by cabamide. The N-doped TiO_2_ coatings were prepared via a dip-coating method on glass substrates (30 × 30 × 2 mm) and then annealed in air at 490 °C for 3 h. The samples were characterized using X-ray diffraction (XRD), scanning electron microscopy (SEM) and UV-vis. The doping rate of N ranged from 0.1 to 0.9 (molar ratio), which caused redshifts to a longer wavelength as seen in the UV-vis analysis. The photocatalytic activity was investigated in terms of the degradation of phenol under both UV light and visible light over 4 h. Under UV light, the degradation rate of phenol ranged from 86% to 94% for all the samples because of the sufficient photon energy from the UV light. Meanwhile, under visible light, a peak appeared at the N-doping rate of 0.5, which had a degrading efficiency that reached 79.2%, and the lowest degradation rate was 32.9%. The SEM, XRD and UV-vis experimental results were consistent with each other.

## 1. Introduction

In recent years, substances that cause water pollution are mainly refractory pollutants in industrial and agricultural wastewater and domestic sewage, such as volatile halogenated hydrocarbons, phenols, nitrobenzene and polycyclic aromatic hydrocarbons. Much effort has been devoted to the photocatalysis field in water treatment to efficiently remove nonbiodegradable compounds and avoid secondary pollution. The photocatalytic reaction transfers an electron from a valance band electron to an empty conduction band by absorbing photon energy equal to or more than the semiconductor band gap. The resulting electron–hole pairs contribute to the degradation reaction of the pollutant. Phenolic compounds are highly toxic and carcinogenic; they are present in wastewater, mainly in effluents from the production of pharmaceuticals, plastics, pesticides, oil and petrochemicals. Phenol was used as a model form of pollution since its molecular structure contains a benzene ring, making it quite stable and difficult to be biodegraded.

Common semiconductor photocatalysts include TiO_2_, WO_3_, ZnO and NiO. In order to improve the photocatalytic degradation activity, many technologies were proposed, such as metal or nonmetal doping, the development of microspheres and coupling semiconductors together. A pure TiO_2_ photocatalyst often has some shortcomings, such as easy recombination of photogenerated electrons and holes, low catalytic efficiency and only responds to UV light, which limits its practical application. To solve these problems, current research focuses on metal/nonmetallic element doping, precious metal deposition and the construction of composite photocatalysts. TiO_2_ of ultrafine nanoparticle size is considered to be the most useful photocatalyst for its excellent properties, such as high-light-conversion efficiency, chemical stability, nontoxic nature and low cost [1,2,3,4,5,6,7,8,9,10,11,12,13].

The band gap energy of TiO_2_ is 3.2 eV (for anatase) and 3.0 eV (for rutile), and the maximum absorption wavelength of TiO_2_ is 387.5 nm (anatase); that is to say, TiO_2_ can only assimilate UV light rather than generate electrons (e^−^) and holes (h^+^), which can subsequently induce redox reactions for the degradation of nonbiodegradable organics in water [14]. Under the irradiation of UV light, electrons are promoted from the valence band to the conduction band of the semiconductor, creating electron–hole pairs, which can cause highly oxidizing hydroxyl and highly reduced superoxide radicals [15]. While the energy of UV light only takes 4–5% of the solar energy, how to enlarge the maximum absorption wavelength of TiO_2_ for visible light and cause TiO_2_ to absorb more visible light has become a research hotspot in recent years [16]. Ion doping, semiconductor complexes and surface photosensitized methods were employed to cause TiO_2_ to be responsive to visible light. Nonmetallic ion doping is one of the widely studied ways of inducing new electronic bands and optical transitions, which involves the inclusion or substitution of a foreign atom, such as nitrogen, sulfur, fluorine or sulfur, that replaces the oxygen atom in the TiO_2_ crystal lattice [17,18,19]. Doping TiO_2_ with nitrogen can create a redshift in the absorption wavelength from UV to the visible range because of the formation of new states inside the TiO_2_ bandgap. This shift could enable photocatalytic reactions to produce a high degradation rate under sunlight illumination [20,21,22,23].

Different nano-TiO_2_ photocatalytic systems in suspension were studied, such as nanoparticles, nanobelts and nanotubes, where TiO_2_ has a larger specific surface area and high absorption of light, and all show increased photocatalytic activity when excited under visible light toward the degradation of different chemical species [24,25,26]. However, difficulties in separation and recycling lead to the smaller possibility of industrial application. Furthermore, the dosage of TiO_2_ is difficult to control, where too little will lead to low photocatalytic efficiency and too much will cause light scattering that influences the absorption of light. Thus, much effort was expended to immobilize photocatalysts in the form of thin films on a stable support to avoid the problems associated with disposing photocatalyst suspensions [27]. Different contents of doped nitrogen have different impacts on the photocatalytic efficiency, for disparities in the replaced oxygen atom can lead to variations in the photocatalyst activity. Frequently used nitrogen sources include urea, triethylamine, ammonia and ethylmethylamine [25].

N-doped TiO_2_ thin films can effectively solve the previous problems by enlarging the maximum absorption wavelength and realizing immobilization together. Nitrogen is doped into TiO_2_ using the sol–gel method, which is easy to operate and the reaction condition is mild. The formation of catalyst needs to go through the procedure of dipping, pulling out, drying, annealing, and chilling [28].

The purpose of this research was to prepare nitrogen-doped TiO_2_ thin films on sheet substrates of different nitrogenous amounts using the sol–gel method. The advantages of this method are that TiO_2_ can be stimulated with visible light, it solves the traditional issue of photocatalytic technology being difficult to control, problems such as high cost and effective components being easily lost are solved, and it greatly improves the possibility of a large-scale practical application in wastewater treatment engineering. We compared the degradation properties of these samples, investigated degradation results and their characterization consequences, analyzed the influence of the amount of nitrogen, and addressed the relationship between the concentration of zymolyte and the reaction time.

## 2. Materials and Methods

### 2.1. Materials and Reagent

Tetrabutyltitanate (CP, 98%), acetylacetone (AR, 98.0%), polyethylene glycol, acetone (AR, 99.5%) and hydrofluoric acid were provided by GuangFu Fine Chemical Research Institution in TianJin. Anhydrous ethanol (AR, 99.7%) and carbamide (AR, 99.5%) were made by ShuangShuang Chemical Co., Ltd., in YanTai, China. Deionized water was created in our lab. The glass substrates used were common commercial glass sheets cut to the needed size.

### 2.2. Preparation of N-Doped TiO_2_ Films

The dip-coating method was utilized to immobilize the films. Precursor solutions for N-doped TiO_2_ coatings were prepared with tetrabutyltitanate, anhydrous ethanol, acetylacetone, carbamide, deionized water and polyethylene glycol (1 wt%). Tetrabutyltitanate (20 mL), anhydrous ethanol (80 mL) and acetylacetone (3 mL) were mixed using magnetic stirring for 20 min, followed by a certain quality of carbamide added into the solution as the donor of nitrogen (the molar ratios of N/Ti used were 0.1, 0.3, 0.5, 0.7 and 0.9). Deionized water (15 mL) was dropwise added at the speed of one drop per second and left to stir for one hour; then, the procedure was finished after the addition of polyethylene glycol (1 mL) to avoid fracturing of the films [28]. Undoped TiO_2_ sol with the same reagents and procedures was also prepared to compare the photocatalytic activity with N-doped coatings. The sol was aged for 24 h at 35 °C and then used to make the coatings.

Commercial glass substrates (30 mm × 30 mm × 2 mm) cleaned with acetone (10 wt%) and eroded with hydrofluoric acid (10 wt%) for 3 h were used as the support of the N-doped TiO_2_ films. The substrates were maintained in the aged sol for 10 min and then pulled out at the rate of 2 cm/s using a dip coater (CZ-4200, Qingdao Zhongrui Intelligent Instrument Co., LTD, Qingdao, China). After the films dried out, the previous process was repeated another three times; the product was then calcined for 3 h at 490 °C. The films were finally even and tight on the glass sheet. The coatings obtained were named TN0, TN1, TN3, TN5, TN7 and TN9 according to the amount of doped nitrogen.

### 2.3. Characterization

After the deposition of the TiO_2_ films onto glass substrates, all of the remaining sol was dried at 80 °C for 20 h in order to obtain dried gels, which were then calcined at 490 °C for 3 h to prepare the powders with the same crystal phase as the coatings. These powders were synthesized to analyze the phase compositions of titania by means of X-ray diffraction (XRD) using a D/Max-2200 Powder X-ray Diffractometer (XRD, D/Max-2200, Nippon Science Corporation, Tokyo, Japan) with Cu-Kα radiation at 40 kV and 20 mA.

Scanning electron microscopy (SEM, SU8220, Hitachi hi-tech, Shanghai, China) was utilized in an air atmosphere to examine the morphological structure and grain size of the films coated on the glass. The vapor has its considerable impact on the dried sample before observation [29].

The UV-vis DRS measurements were recorded at room temperature for the dry-pressed disk samples using a UV-3600 UV-vis spectrophotometer (UV-3600 UV-vis, Shimadazu, Tokyo, Japan) equipped with an integrating sphere assembly within the range of 300–900 nm.

### 2.4. Measurement of Photocatalytic Activity

The experiments were carried out with the initial concentration of phenol equal to 10 mg/L at an ambient temperature (approximately 25 °C) and pressure. The photoreactor used an acrylic cuboid static opaque chamber (700 mm × 450 mm × 250 mm) equipped with a thermocouple to monitor the temperature during irradiation. The UV source was supplied by 6 ultraviolet tubes (20 W), which had a dominant wavelength of 254 nm. As for the visible light source, 6 ordinary fluorescent lamps (20 W) were employed to produce the longer wavelength light.

Each beaker contained 8 pieces of glass sheet supporting N-doped TiO_2_ coatings of the same amount of doped nitrogen. The illuminant was about 15 cm from the bottom of the solution. The system was left in the dark for 30 min until reaching phenol adsorption equilibrium, and then a photocatalytic reaction was carried out under UV light or visible light. The samples were taken from the reactor for analysis every 30 min, where the samples were placed in a 2 cm quartz dish and the remaining concentrations were analyzed using 4-AAP extraction spectrophotometry and UV-vis spectrophotometry (Photo Lab 6600 UV–Vis, WTW, Munich, Germany) at 510 nm. The photocatalysis reaction lasted for 4 h.

## 3. Results and Discussion

### 3.1. XRD Analysis

Figure 1 shows the XRD patterns of the six powdery samples. For all samples, it can be observed that where the 2θ was 25.4° (101), the diffraction peak was especially distinct, and at 2θ = 30.7° (121), the relative intensity is quite small, which means that the anatase phase was dominant and the rutile phase was hardly existing. In addition, other characteristic peaks (2θ = 25.34°, 48.11° and 44.49°) were all accordant with JPDS-21-1272 [30] (anatase standard card). It can be confirmed that N-doped TiO_2_ mainly existed in the form of anatase. With the increase in the N doping amount, the sample pattern almost did not change, indicating that N doping had little effect on the crystal structure. The difference being all samples on the 101 crystal plane may have been caused by the grinding of the samples with different N contents.

### 3.2. SEM Analysis

Figure 2 shows the SEM photographs of a clean glass substrate and N-doped TiO_2_ films. The glass substrate after being washed with acetone and eroded with hydrofluoric acid was quite clean and had an even roughness with a great transmission of light. A rough surface can increase the capacity of the load [27]. The microstructures of the six samples had no obvious differences, which indicated that the quantity of carbamide had little impact on the surface of the coatings. Figure 2d, e demonstrate that the films made using the sol–gel method were porous and the granules dispersed uniformly with particle sizes ranging from 30 nm to 80 nm.

### 3.3. UV-Vis DRS Analysis

UV-vis DRS results are shown in Figure 3. The maximum absorption wavelength of TN0 was about 380 nm, indicating the main components of samples were naked titanium dioxide with little impurities. Different amounts of N-doping replaced oxygen atoms in the TiO_2_ crystal lattice with nitrogen atoms to different degrees, which was also the reason for different degrees of redshift when compared with the maximum absorption wavelength of TN0. It is worth noting that the nitrogen doping amount did not follow a “the more, the better” pattern within a certain range since the maximal redshift occurred for TN5, not TN9. Different amounts of N-doping can reduce the bandwidth of TiO_2_, enhance the transfer of electrons from the valence band to the conduction band and improve the photocatalytic rate. However, when the amount of N-doping is too much, the nitrogen atom will become the center of electron recombination and accelerate the recombination rate of electrons and holes, thus affecting the photocatalytic rate [31,32].

### 3.4. Photocatalytic Activity under UV and Visible Light Irradiation

#### 3.4.1. Photocatalytic Activity under UV Light Irradiation

With regard to absorption, the dark test showed that the variation of phenol concentration caused by absorption was lower than 3%, which meant that the absorption had little influence on the degradation ratio of the photocatalytic procedure.

The variation of each sample over time is shown in Figure 4. The phenol concentration at 0 min was measured just after absorption. It can be observed that the rank of the six samples regarding photocatalytic activity was TN5 > TN3 > TN7 > TN1 > TN9 > TN0, i.e., TN5 had the highest degradation over the others during the same period. The degradation ratios were 93.77%, 91.32%, 90.82%, 88.89%, 85.73% and 80.10%, respectively. The maximum absorption wavelength of the five N-doped samples were all redshifted to visible light to different degrees, while the photocatalytic activities of the five samples were also enhanced, which should have been caused by the doping nitrogen leading to the energy structure of titanium dioxide changed; that is to say, the optical energy band gap of TiO_2_ diminished. A reduction in the optical energy band gap will enhance the transfer of electrons from the valence band to the conduction band under visible light, which may have been the reason for the better relative performance of the N-doped TiO_2_ [33]. The kinetics of phenol removal followed the Langmuir–Hinshelwood kinetic equation [34]:(1)$$R=d_{c}/d_{t}=kK_{c}/\left(1+K_{c}\right)$$
where R represents the reaction rate, c is the concentration of the substrate at the time, k is the reaction rate constant and $K_{C}$ is the adsorption constant. When the concentration was very low, $K_{c}$ << 1; therefore, the relevant equation turned out to be
(2)$$\ln\left(c_{0}/c\right)=k_{t}+b$$
where $c_{0}$ is the initial concentration, t is the reaction time and $k_{t}$ is the apparent reaction rate constant. Figure 5 shows the kinetics of the six samples. The apparent reaction rate constants ranged from 0.00614 to 0.01048.

#### 3.4.2. Photocatalytic Activity under Visible Light Irradiation

The absorption under visible light in the dark period made little difference compared with UV light, while the variation in phenol concentration during illumination time made a significant difference.

It can be observed from Figure 6 that the degradation rates of the N-doped samples were obviously better than that of the naked TiO_2_ (TN0). TN5 still possessed the fastest degradation rate, followed by TN3, TN7, TN1 and TN9; that is to say, the best degradation efficiency appeared at the point where the molar ratio of N/Ti was 0.5, rather than the more nitrogen doping, the higher the degradation rate. The degradation results were also coincident with the UV-vis DRS results, where TN5 had the greatest redshift.

The kinetics of phenol removal under visible light also followed the Langmuir–Hinshelwood kinetic equation (the kinetic Equation (1) in 3.4.1), as Figure 7 shows. The apparent reaction rate constants ranged from 0.00156 to 0.04893.

## 4. Conclusions

N-doped TiO_2_ thin films were successfully immobilized on commercial glass substrates via the sol–gel method starting from tetrabutyltitanate dissolving in anhydrous ethanol as a precursor. The formulation of sol and an annealing temperature of 490 °C were optimal, as seen by the highly uniform lattice structure that was mainly constituted of anatase and the TiO_2_ granules being evenly distributed with ultrafine nano-particle sizes ranging from 30 to 80 nm. The surface morphology of the coating was basically unaffected by different nitrogen contents. Under the same experimental conditions, the degradation efficiency of phenol in the experimental group under visible light irradiation reached about 90% of that under UV light, indicating that N-doping caused the optical energy band gap of TiO_2_ to diminish; therefore, the maximum absorption wavelength had obvious redshifts, leading to the doped films having more efficient photocatalytic activity, both under UV light and visible light. The optimal photocatalytic efficiency was realized at an N-doping ratio of 0.5, rather than the more dopant, the better the photocatalytic efficiency since an excess N-doping ratio leads to an increased recombination ratio of electrons and holes, which reduces the photon utilization factor. The pollution absorption ability of the TiO_2_ and glass sheet was quite feeble; thus, the kinetics of degradation followed the Langmuir–Hinshelwood kinetic equation, which describes first-order reaction kinetics.

## Figures and Tables

**Figure 1 ijerph-19-15721-f001:**
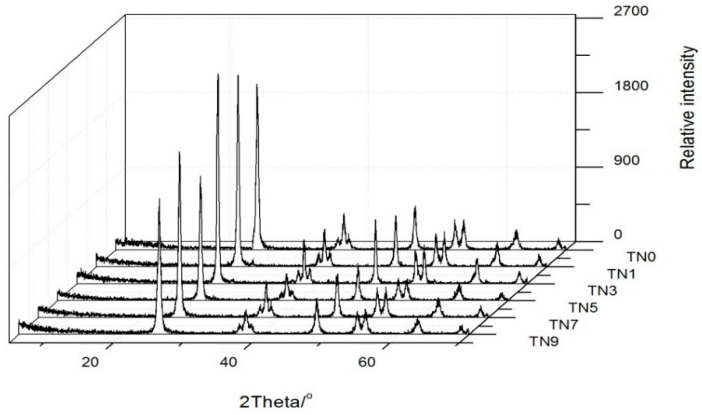
XRD patterns of TN0, TN1. TN3, TN5, TN7 and TN9.

**Figure 2 ijerph-19-15721-f002:**
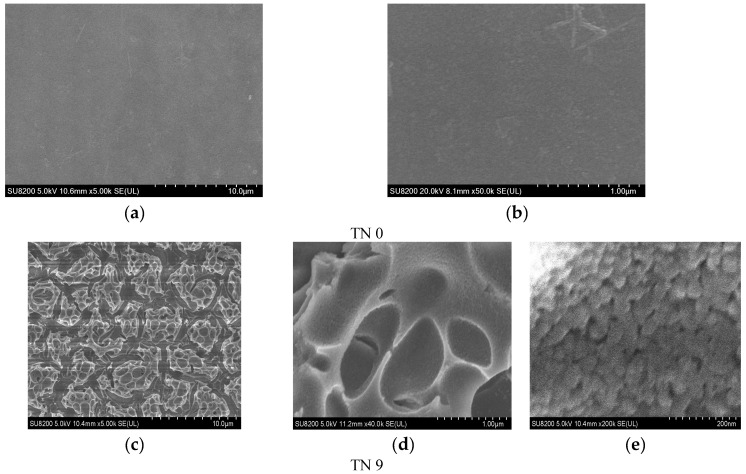
SEM photographs of a clean glass substrate (**a**,**b**) (TN0) and SEM photographs of sample TN9 (**c**–**e**).

**Figure 3 ijerph-19-15721-f003:**
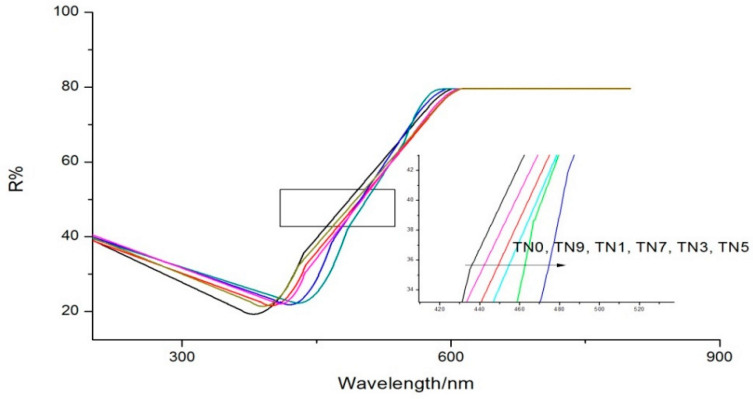
UV-vis DRS of TN0, TN1, TN3, TN5, TN7 and TN9.

**Figure 4 ijerph-19-15721-f004:**
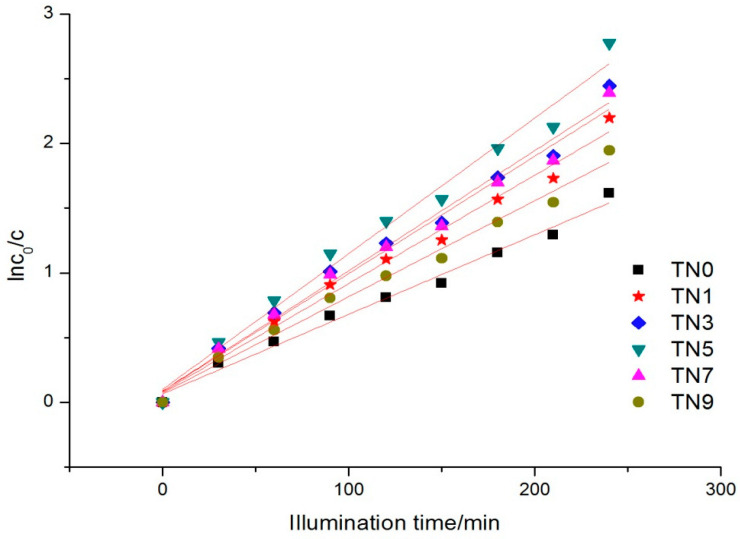
Phenol concentration trend under UV light.

**Figure 5 ijerph-19-15721-f005:**
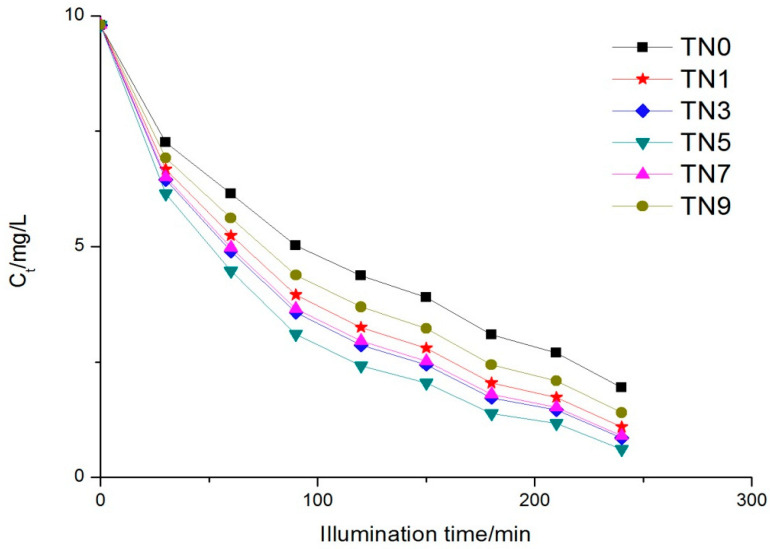
Reaction kinetics trend under UV light.

**Figure 6 ijerph-19-15721-f006:**
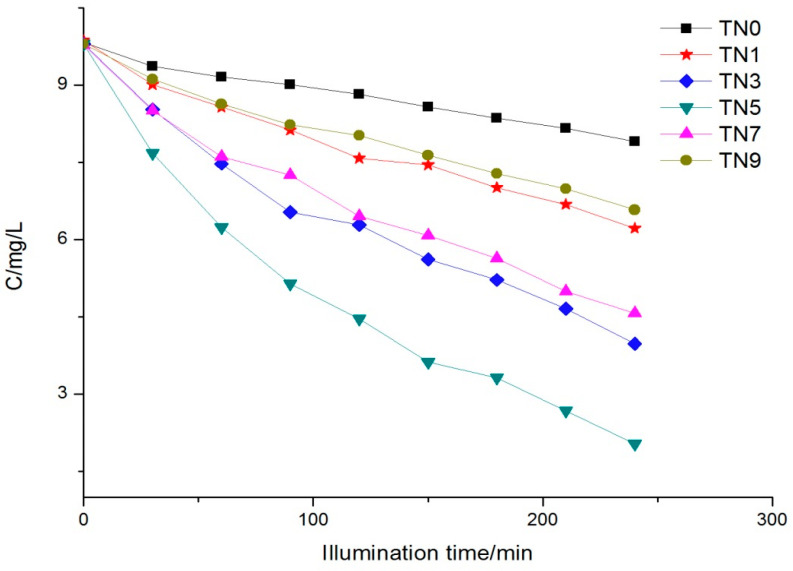
Phenol concentration trend under visible light.

**Figure 7 ijerph-19-15721-f007:**
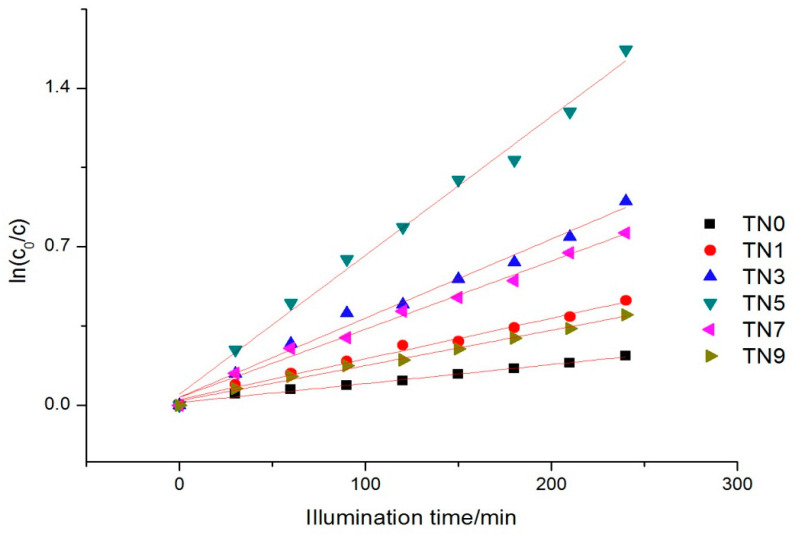
Reaction kinetics trend under visible light.

## Data Availability

Not applicable.

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
