# Peer review of "The Preparation of N-Doped Titanium Dioxide Films and Their Degradation of Organic Pollutants"

_ijerph, 2022, doi:10.3390/ijerph192315721_

Round 1

Reviewer 1 Report

The manuscript discusses how N-doped titanium dioxide film can break down organic pollutants through its photocatalytic activity. It is an important field of research. However, it needs a major revision.

1–What is the novelty of this work, and how is it different from most photocatalytic TiO2 studies published previously?

2-The authors need to supplement the necessary characterizations to illustrate the successful preparation of N-doped TiO2, such as FT-IR and XPS. Also, analysis (like FTIR) should be done before and after photodegradation to determine the stability of the photocatalysts.

3- The authors stated in the XRD section, " Six patterns are almost the same, which attests that the doping of carbamide has no impact on the crystalline phase. Differences among all the samples at 101 crystallographic plane are caused by grinding". Please clarify this.

4-The authors stated in the UV-vis DRS result, " N-doped samples show obvious difference compared with TN0 red shift of the maximum absorption wavelength caused by nitrogen atom replacing oxygen atom in the crystal lattice". This disagrees with the XRD interpretation. Also, I suggest changing from N-doped TiO2 to Nx-doped TiO2-x.

5-In Figure 2, the authors should use the same magnification to compare the substrate image with the materials.

6- The authors need to determine the energy band in Figure 3, clarify the transfer direction of photogenerated carriers, and propose a suitable photocatalytic degradation mechanism.

7- Please check Figures 4 and 5 with their caption. I suggest re-drawing them using lines and dots to avoid confusion.

8- The authors should list comparisons between N-doped TiO2 material and similar or other catalysts to show the innovation of this catalyst. Some recent papers involving photocatalytic environmental remediation can be referenced in the manuscript.

-Journal of Molecular Structure, 1250 (2022) 131800.

- Journal of Environmental Management 258 (2020) 110043.

-Environ Sci Pollut Res 29, 56845–56862 (2022).

-Materials Chemistry and Physics 242 (2020) 122520.

Reviewer 2 Report

Point 1.

The Introduction is compact and well-written, but I would add some case studies related to water treatment to highlight the importance and applicability of the study.  

Point 2. I’d add more explanation why the phenol was used as model pollution (line 70-72)

Point 3 line 130-134. Not quite clear, please add some explanation

Point 4. line 177-182  The equations are poorly formatted and explained, please revise it

Point 5. Figure should not be separated from the text; it should appear closer where these are referenced. Some Figure caption should be more precisely stated

Point 6. The discussion of the study is missing, please compare your results with literature data and/or put your result in a context of other studies.

Point 7. Please proofread your text, there are some typos and use formal technical language

Round 2

Reviewer 1 Report

The manuscript was improved and can be published in the International Journal of Environmental Research and Public Health.

Reviewer 2 Report

Thank you for the revised paper, I accept the modificiations.